# Properties and Emissions of Three-Layer Particleboards Manufactured with Mixtures of Wood Chips and Partially Liquefied Bark

**DOI:** 10.3390/ma16051855

**Published:** 2023-02-24

**Authors:** Wen Jiang, Stergios Adamopoulos, Reza Hosseinpourpia, Thomas Walther, Sergej Medved

**Affiliations:** 1Department of Technical Physics, University of Eastern Finland, P.O. Box 1627, 70211 Kuopio, Finland; 2Department of Forest Biomaterials and Technology, Swedish University of Agricultural Sciences, Vallvägen 9C, 75007 Uppsala, Sweden; 3Department of Forestry and Wood Technology, Linnaeus University, Lückligs Plats 1, 35195 Växjö, Sweden; 4College of Forest Resources and Environmental Science, Michigan Technological University, Houghton, MI 49931, USA; 5IKEA Industry AB, Skrivaregatan 5, 21532 Malmö, Sweden; 6Department of Wood Science and Technology, Biotechnical Faculty, University of Ljubljana, Rožna Dolina C VIII/34, SI-1000 Ljubljana, Slovenia

**Keywords:** bark residues, FTIR, liquefaction, particleboard, SEM, formaldehyde emissions, VOC emissions

## Abstract

Partial liquefaction of residual biomass shows good potential for developing new materials suitable for making bio-based composites. Three-layer particleboards were produced by replacing virgin wood particles with partially liquefied bark (PLB) in the core or surface layers. PLB was prepared by the acid-catalyzed liquefaction of industrial bark residues in polyhydric alcohol. The chemical and microscopic structure of bark and residues after liquefaction were evaluated by means of Fourier Transform Infrared Spectroscopy (FTIR) and Scanning Electron Microscopy (SEM), while the particleboards were tested for their mechanical and water-related properties, as well as their emission profiles. Through a partial liquefaction process, some FTIR absorption peaks of the bark residues were lower than those of raw bark, indicating hydrolysis of chemical compounds. The surface morphology of bark did not change considerably after partial liquefaction. Particleboards with PLB in the core layers showed overall lower densities and mechanical properties (modulus of elasticity, modulus of rupture, and internal bond strength), and were less water-resistant as compared to the ones with PLB used in the surface layers. Formaldehyde emissions from the particleboards were 0.284–0.382 mg/m^2^·h, and thus, below the E1 class limit required by European Standard EN 13986:2004. The major emissions of volatile organic compounds (VOCs) were carboxylic acids as oxidization and degradation products from hemicelluloses and lignin. The application of PLB in three-layer particleboards is more challenging than in single-layer boards as PLB has different effects on the core and surface layers.

## 1. Introduction

Particleboard is an engineered panel product made from wood flakes or shavings bonded by synthetic adhesives such as urea–formaldehyde (UF), phenol–formaldehyde (PF), melamine–formaldehyde (MF), and isocyanate [1,2]. This panel product finds wide applications in furniture, construction, and packaging due to its low cost and availability on the market in a wide range of dimensions [1,3]. The global production of particleboard increased by 61.7% from 64.3 million m^3^ in 2000 to 104 million m^3^ in 2021, and the growth is expected to continue rising [4]. A large variety of lignocellulose-based resources serve as raw materials, such as wood, natural fibers, forest-based and agro-industrial side-streams and residues, and recycled wood [5,6,7]. Research on particleboards is now focused on finding homogenous non-wood feedstock and bio-based adhesives with low environmental impact [1]. Bark, as a side-stream from forest-based industry, is a very useful material for producing composites (bark-wood-based composites, binderless particleboards, insulation panels, etc.) or extracting tannins, which can be used in wood adhesive formulations, preservatives or water repellents [8,9,10].

Lignocellulosic materials can be liquefied in different organic solvents with the use of acidic catalysts and the obtained products can be used in adhesive systems, such as urea–formaldehyde (UF), melamine–formaldehyde (MF), phenol–formaldehyde (PF), melamine–urea–formaldehyde (MUF), polyurethane and epoxy [11,12]. The basic mechanism of the acid-catalyzed liquefaction of lignocellulosic biomass involves solvolytic reactions under acidic conditions to form smaller fragments, which further react with themselves or the solvents to form higher molecular weight fragments or solvent-derived compounds [11,13,14]. Adhesives based on liquefied biomass are suitable for bonding wood or manufacturing particleboards and other types of wood-based panels. Hassan et al. [15] liquefied pine wood in phenol-based solvent with a phenol-to-wood ratio of 70:30, 65:35 and 60:40 under 160–165 °C for 1 h, and then synthesized PF adhesives by reacting liquefied wood (LW) with formaldehyde. Three-layer particleboards bonded by the LW-based PF adhesives showed better water resistance, comparable bending strength and lower free-formaldehyde emissions but lower internal bond (IB) strength than the controls. Antonović et al. [16] reported that particleboards bonded by UF adhesives mixed with LW had significantly reduced formaldehyde emissions than UF-bonded controls; however, LW did not contribute to the adhesion. In contrast, adhesives synthesized from LW and formaldehyde led to enhanced mechanical properties of particleboards as well as to higher free-formaldehyde emissions. Kunaver et al. [17] reported that the addition of 50% LW to MF and MUF adhesives reduced formaldehyde emission three-layer particleboards by up to 40%, hence, meeting the European standards. Janiszewska [18] liquefied bark in glycerin and propylene glycol mixtures under 120 °C for 2 h and used the liquefied bark (LB) to replace 20% of MUF adhesives for bonding three-layer particleboards. The substitution of MUF with LB decreased static bending and tensile strength mainly due to the acidity of LB. Zhang et al. [19] applied liquefied soybean protein when producing adhesives for bonding particleboards. As reported, the liquefied soybean protein-based adhesives had low viscosity, good bonding strength and good processability.

Different from liquefied biomass functioning as adhesives or used for modifying adhesives, Jiang and co-authors [20,21] first reported on the partial liquefaction of bark and used partially liquefied bark (PLB) as a furnish material with binding ability for producing particleboards. PLB is a crude partial liquefaction product containing unreacted solvents and catalysts, intermediate products and solid bark residues [20]. Acid-catalyzed liquefaction has similar operating temperatures to those applied in the pressing procedure for manufacturing particleboards; therefore, it was previously hypothesized that the partially liquefied biomass can be reactivated during the hot pressing of boards [20,21,22], thus, full in situ liquefaction may occur. Jiang and co-authors [20,21] developed single-layer particleboards using partially liquefied bark (PLB). They investigated the influence of bark fraction sizes and loading levels of PLB on the physical and mechanical properties of particleboards. PLB was prepared by the liquefaction of pine bark in ethylene glycol with sulfuric acid as a catalyst under 180 °C for 30 min. Single-layer particleboards were successfully produced from PLB and wood particles without adhesives; however, those particleboard panels exhibited low mechanical strength. Introducing PLB up to 9.1% to MUF-bonded particleboards improved the mechanical properties, and the higher level of PLB in particleboards reduced thickness swelling and water absorption. This indicated that PLB acted as an excellent water repellent agent for single-layer particleboards. As PLB introduced unreacted solvents and acid catalysts to the surface of wood particles, chemical bonding was formed between wood and PLB with a densified area observed by SEM [20]. A similar study was reported by Choowang and Luengchavanon [22]; they used a suspension of oil palm trunk and citric acid to bond two rubberwood veneers by hot pressing at 160–200 °C for 3–7 min. Citric acid acted as a cross-linking agent and catalyst for bonding the rubberwood veneers while the acid hydrolysis of carbohydrate polymers in the cell walls of the oil palm trunk negatively influenced bonding. Nitu et al. [23] used a citric acid–glycerol mixture as a natural binder for bonding the jute stick particleboard, and presented cross-linked networks with enhanced properties during particleboard production; this is a similar idea—using unreacted solvents and catalysts in the PLB to form chemical bonding between PLB and wood particles. 

The previous studies showed great potential to produce single-layer particleboards with PLB and wood particles [20,21]. It was found that the liquid phase of PLB acted as a heat transfer medium, enhancing the bonding between PLB and wood particles. This study aimed to apply PLB in three-layer particleboards for replacing part of virgin wood particles and reducing the use of commercial synthetic adhesives. It is hypothesized that the distribution of PLB in the panel layers, e.g., surface and core layers, has a different effect on the activation of PLB, and thus, the resulting panel performances. The chemical and micro-structural changes of bark particles due to the partial liquefaction process were studied with FTIR and SEM, and the particleboard panels were analyzed by means of their physical and mechanical properties as well as their emissions (formaldehyde and VOCs).

## 2. Materials and Methods

### 2.1. Materials

Norway spruce (*Picea abies* (L.) H. Karst.) bark used for liquefaction was collected from a sawmill (JG Anderson’s Söner AB, Linneryd, Sweden). The bark residues were oven-dried at 40 °C overnight before milling down to the size of 2 mm. Chemicals used for partial liquefaction were ethylene glycol (VWR International BVBA, Leuven, Belgium) and sulfuric acid with a purity of 95% (VWR International S.A.S., Fontenay-sous-Bois, France). Industrial wood chips (mixtures of Scots pine (*Pinus sylvestris* L.) and Norway spruce), MUF (solid content of 68%) and ammonium sulfate (solid content of 50%) were kindly provided by IKEA industry AB (Hultsfred, Sweden). 

For characterizing the bark particles after partial liquefaction, 1,4-dioxane (VWR International S.A.A, Fontenay-sous-Bois, France) and acetone (Supelc^®^, Merk Life Science UK Limited, Gillingham, UK) were used to remove the remaining solvents and intermediate products from PLB.

### 2.2. Partial Liquefaction Process

Spruce bark was partially liquefied following the method described previously by Jiang et al. [20]. In brief, oven-dried (103 °C for 24 h) milled bark was mixed with ethylene glycol as a solvent at a solvent-to-biomass mass ratio of 3:1, and sulfuric acid (3% based on solvent *w/w*) as a catalyst in a three-neck glass reactor. The reactor was immersed in an oil bath and heated at 180 °C for 30 min with constant stirring. The reaction was then stopped by moving the reactor from the oil bath and keeping it under the fume hood until the reactor was cooled to room temperature. Afterwards, PLB was collected and transferred to a clean beaker for further use. 

### 2.3. Characterization of Partially Liquefied Bark

Two samples (PLB residue and polyol) were prepared from the PLB for analysis of the changes in the bark structure. Crude PLB was diluted in 1,4-dioxane: water (4:1) and centrifuged to remove solid residues. The washing procedure was repeated twice by using acetone to obtain clean PLB residue, which had no unreacted solvent, catalyst and liquid intermediate products. The PLB residue was then oven-dried at 103 °C for 24 h. The PLB polyol was obtained from the dioxane and acetone soluble solutions followed by a rotary evaporation of dioxane and acetone. 

The chemical structure of raw bark, PLB polyol and PLB residue were characterized with a Fourier Transform Infrared (FTIR) Spectrometer (Perkin Elmer, Bruker, Karlsruhe, Germany) equipped with a versatile high throughput ZnSe ATR crystal. The analysis was performed by having 32 scans with a resolution of 4 cm^−1^ and a wavelength range from 4000 to 600 cm^−1^.

Changes in bark structure after partial liquefaction were observed under a Scanning Electron Microscopy (SEM). Raw bark and PLB residue were oven-dried at 60 °C for 4 h. Samples were mounted on aluminum stubs and were coated with gold using an Emitech K550X sputter coater (Quorum Technologies Ltd., Ashford, UK). Prepared samples were then observed using a Philips XL30 ESEM (Philips, Eindhoven, The Netherlands) with a voltage of 15 kV.

### 2.4. Three-Layer Particleboard Production

Two groups of particleboards, groups A and B, with a target density of 620 kg/m^3^ and thickness of 12 mm were prepared using 9.1% of PLB and 8–12% of MUF adhesives in the core layers (CL) or surface layers (SL), respectively, as detailed in Table 1. Reference board C* was produced with 100% wood particles in both surface and core layers as a standard industry board. Control C* was later used to compare with other boards in two different groups. Groups A and B had one extra control board and four PLB-based particleboards. C-A* and C-B* were reference boards for groups A and B, respectively, by having a similar content of solid materials to other boards in the same group. 

### 2.5. Particleboard Testing

Mechanical and physical properties of particleboards were evaluated according to the standards: moduli of elasticity (MOE) and rupture (MOR) in static bending (BS EN 310:1993 [24]), 6 replicates per board; internal bond (IB) strength (BS EN 319:1993 [25]), 8 replicates per board; thickness (EN 324-1:1993 [26]), 8 replicates per board; density (BS EN 323:1993 [27]), 8 replicates per board; thickness swelling (TS) and water absorption (WA) (BS EN 317:1993 [28]), 8 replicates per board. The mechanical tests were performed using a universal testing machine (MTS Exceed E43-10 kN, MTS Systems Corporation, MN, USA). The density profile of the particleboards was determined using a DAX 6000 X-ray microdensitometer (GreCon GmbH, Alfeld, Germany) with 8 samples of dimensions 50 × 50 mm^2^. 

Formaldehyde emissions of the particleboards were determined by a GA-6000 gas analyzer (GreCon GmbH, Alfeld, Germany) according to the standard BS EN 717-2:1994 [29]. The VOCs were determined by the chamber method according to standards BS EN ISO 1600-9:2006 [30] and ISO 16000-6:2011 [31]. Eight specimens were measured for each particleboard type. 

### 2.6. Statistical Analysis

The software SPSS Statistics 25.0 (IBM Corp., Armonk, NY, USA) was used for statistical analysis. The mechanical and physical results were evaluated by analysis of variance (One-way ANOVA) at a 95% confidence interval (*p* < 0.05). The statistical differences between mean values were assessed by the Tukey test.

## 3. Results and Discussions

### 3.1. Characterization of PLB Polyol and Residue

Changes in the chemical structure of the bark after partial liquefaction were characterized by FTIR (Figure 1). Raw bark and PLB residue did not exhibit a distinctly different chemical structure, which suggested that bark had not been liquefied to a great extent. A strong absorption peak at 3340 cm^−1^ corresponded to the stretching vibration of -OH bonds in carbohydrates and lignin [21,32]. The intensity of the -OH vibration peak decreased in oven-dried PLB residue when compared to raw bark, indicating that the hydroxyl groups in the raw bark were released through partial liquefaction [33]. This was also supported by the observation of a significantly increased -OH peak in PLB polyol attributed by the presence of ethylene glycol and its derivatives. Typical absorption bands of ethylene glycol were observed in PLB polyol as follows: strong OH stretching at 3300 cm^−1^; symmetric and asymmetric CH stretching vibrations at 2940 and 2877 cm^−1^; weak bands corresponding to CH_2_ and CH_3_ bending between 1455–1205 cm^−1^; and strong bands at 1085, 1035 and 880 cm^−1^ corresponding to functional groups, namely, C-O stretching, C-C-O asymmetric stretching and C-C-O symmetric stretching [21,34,35]. The only difference between the spectra of PLB polyol and ethylene glycol was found at 1706 cm^−1^ corresponding to C=O stretching, which indicated the formation of a small amount of carboxylic acid as intermediate products. The other peaks of PLB residue decreased at 2911 cm^−1^ corresponding to -OH stretching, at 1609 cm^−1^ corresponding to C=O stretch vibration, and at 1028 cm^−1^ corresponding to -CO stretching when compared to those of raw bark. Such changes confirmed the thermal degradation of the main chemical compositions of bark after partial liquefaction. The SEM images (Figure 2) showed no obvious structural changes in the bark particles before and after partial liquefaction. However, some granules on the bark surface were removed, which could be wood residues left from debarking. The above observations suggested non-noticeable structural and chemical changes of bark after partial liquefaction. 

### 3.2. Physical Properties of Particleboards

The thickness and density of the three-layer particleboards are given in Table 2. The examined particleboards had an average density in the range of 566.2–641.6 kg/m^3^. There was no statistically significant difference between the density of the boards in groups A and B. However, the thickness of the boards in groups A and B had statistically significant differences when using different amounts of MUF for bonding. Considering the evaporation of chemicals in PLB during hot pressing, PLB-C and PLB-S boards definitely had a lower solid content when compared to control C*. Therefore, C-A* and C-B* were considered as additional reference boards for matching similar solid content in the particleboards and for the comparison. All boards containing PLB had a higher thickness springback than the controls, i.e., when comparing PLB-C to C-A* and even C* boards, and PLB-S to C-B* boards. This was also due to the release of vapor from the unreacted solvent and intermediate liquid products from the PLB [36]. Boards in group A had relatively higher thicknesses and lower density than those in group B due to the fact that more wood particles were replaced by PLB in the CL than in the SL. Board PLB-C-III had the lowest density among all the board types. This could be attributed to the smaller amount of MUF adhesives used when compared with other boards in group A. 

Density profile analysis was performed on particleboard samples to show the distribution of the particles. Figure 3 presents the average density profile of each board type. A density profile is usually used for the prediction of the internal bond of panel products by relating to the minimum density [37]. As shown in Table 3, the difference between the mean minimum density of each board type is very small in group B when the formulations of particles were the same in the core layers. However, when using PLB in the core layers (group A), board PLB-C-III had a considerably lower mean minimum density than the other boards, while board PLB-C-I had the highest mean minimum density.

The results of IB strength (Figure 4b) showed the highest value in board PLB-C-I and the lowest in board PLB-C-III among the PLB-C boards while PLB-S boards showed non-statistically significant results in IB strength. Similar results were observed from Table 3 for the medium density of the boards, which verified the validity of a vertical density profile as an efficient method for predicting IB strength. As reported previously [38,39,40], the density profile also correlates with other mechanical properties of particleboards such as MOR and MOE. The bending strength can be predicted by observing the shape of the density profile. Typical U-shaped density profiles were observed for all board types, with the highest mean density in the surface layers and the lowest mean density in the core layers. Thus, the bending properties could not correlate with the density profile in this work since the densities of the boards within the same groups were not statistically different.

### 3.3. Mechanical and Water-Related Properties of Particleboards

The results of mechanical properties are presented in Figure 4. The evaporation of chemicals from PLB (of 41% solid content as reported previously [20]) during hot pressing had a direct effect on the overall mechanical properties, as C* had higher MOE, MOR and IB than all other boards. In group A, MOE, MOR and IB of board PLB-C-I were not statistically significantly different from those of control C-A*, which means that density loss from PLB during hot pressing was as expected. However, in group B, PLB-S-I had significantly lower MOE strength but no statistically different MOR and IB strength when compared to C-B*. This indicated that more than 5% of density loss has occurred. 

When reducing the MUF content from 12% to 10%, MOE and MOR of board PLB-C-II were surprisingly higher than C-A* and PLC-B-I, and were not significantly lower than control C*. Previous studies showed that a limited amount of PLB could enhance the mechanical properties of single-layer particleboards bonded by the same content of MUF adhesives [20,21]. This study suggested that reducing MUF content by 2% could lead to a much better heat transfer rate for PLB, thus resulting in improved static bending strength when using PLB in the core layer. However, the further reduction in MUF content to 8% (PLB-C-III) has led to decreased MOE and MOR when compared to board PLB-C-II. A significant decrease was observed for IB strength in group A when reducing the MUF amount of MUF adhesives from 12% to 8%. IB strength has been reported to be highly correlated with board density [41]. 

In group B, PLB-S particleboards did not have significantly different MOR and IB strength when compared to C* and C-B*, even though the MUF content in the surface layer was reduced from 12% to 8%. The differences between the MOE strength of PLB-S-I, PLB-S-II and PLB-S-III were also not significant. 

The above results indicated that PLB had a different impact on the mechanical properties when used in the surface or core layers. The heating transfer ability of PLB was affected by the amount of MUF adhesives. PLB and the MUF adhesive amount seemed to have less negative impact on the mechanical strength of the particleboards when PLB was applied in the SL as compared to CL.

Results of the TS and WA of particleboards are presented in Table 4. Replacing wood with PLB in the CL did not cause statistically significant differences on the 2 h and 24 h TS—only the board PLB-C-III had a significantly higher 2 h and 24 h WA than the other boards in group A. This indicated that a replacement of wood particles with 9.1% PLB and at the presence of at least 10% MUF adhesive content in the CL can ensure comparable water resistance to the control boards. The decreased amount of MUF adhesive from 12% to 10% in the SL caused an increase in TS and WA both after 2 h and 24 h of soaking time in water; however, the difference between particleboards with 10% and 8% MUF in the SL was not significant. In group B, all PLB-S boards had higher density and TS than reference board C-B*. This is due to the fact that particleboards with higher densities are generally characterized by greater dimensional changes [3].

### 3.4. Formaldehyde and VOCs Emissions

Table 5 presents the formaldehyde emission values of different particleboards made from PLB and wood particles bonded with MUF adhesive. Particleboards except PLB-C-I and PLB-S-II had lower formaldehyde emissions than the control board C*, mainly due to the replacement of furnish materials by PLB that resulted in reduced MUF adhesive contents. No obvious decline in formaldehyde emission values with decreasing MUF content could be observed in both groups A and group B. This can be explained by the fact that both wood and bark can emit formaldehyde under thermal treatment [42]. Partial liquefaction, as a thermochemical treatment method, accelerated the release of formaldehyde from the chemical composition in the bark. During the acid-catalyzed liquefaction process, polysaccharides are hydrolyzed and transformed to hexoses and hydroxylmethylfurfural and the subsequent disproportionation to furfural and formaldehydes [13,43,44]. Additionally, the treatment of lignin with acid leads to the liberation of formaldehyde [42,44]. Formaldehyde might have been released from the wood particles during the particleboard production as PLB brought unreacted acid catalyst and acidic intermediates. All the particleboards produced in this work did not exceed the limit by European standards and had relatively lower formaldehyde emissions than the literature. The formaldehyde emissions of all the particleboards can be classified as E1 with a release lower than 3.5 mg/m^2^·h according to the respective European standards for wood-based panels used in construction [45]. Akkuş et al. [46] reported a range of formaldehyde emissions of 1.13–2.12 mg/m^2^·h on 18 mm-thick single-layer particleboards. Salem et al. [47] produced 16, 18 and 19 mm-thick single-layer particleboards and reported a formaldehyde release of 0.44–1.68 mg/m^2^·h while formaldehyde emissions increased with an increasing thickness of particleboards. 

Besides formaldehyde, other types of aldehydes and VOCs are also of great importance for particleboards as indoor furniture and decorative materials. The concentrations of the different categories of VOCs from the particleboards after 72 h and 28 days of tests are illustrated in Figure 5. The names and concentrations of the VOCs can be found in Appendix A (Table A1 and Table A2) for the 72 h and 28 d chamber test. The results showed that PLB significantly affected the total volatile organic compound (TVOC) emissions whether it was used in the SL or CL. The TVOCs emissions of particleboards with PLB in the CL were, respectively, higher than those with PLB in the SL and were still detected at a high level after 28 days This suggested that a large content of fluids from PLB remained in the CL and liberated in the process of time, although the hot pressing during the particleboard production helped the liberation of fluids from PLB. Aldehydes including formaldehyde, acetaldehyde, hexanal, benzaldehyde, octanal, nonanal and furfural were steadily increased by reducing the MUF amount. These aldehydes were mainly degradation products of the secondary components of wood or bark [48]. Alcohols and esters were only detected in the particleboards containing PLB, and the contents were the remaining liquefaction solvent (ethylene glycol) and its glycol esters (listed in Table A1 and Table A2). Other odorous VOCs, such as alkanes, alkenes and aromatics, were detected in a comparably low quantity. Among the VOCs, carboxylic acids emitted in large amounts for all particleboards were much higher in PLB-based particleboards than in the controls. It indicated that carboxylic acid as an oxidization and degradation product from hemicelluloses and lignin was formed either during partial liquefaction or particleboard production [49,50]. Emissions of acetic acid and propionic acid were comparably higher than other acids for all the boards and were increased by PLB, and the latter can be metabolized by the liver of humans according to Al-Lahham et al. [51]. As reported by Ernstgård et al. [52], acetic acid can cause nasal irritation at an exposure amount over 10 ppm (10,000 µg/m^3^) or 3 ppm (3000 µg/m^3^) for 10 s, which are above the highest amount of acetic acid tested in this study. Among all tested VOCs, styrene, toluene, xylene and formaldehyde are the ones with short-term adverse health effects, such as eye, nose, throat and skin irritation, and long-term adverse health effects such as a loss of senses (color discrimination, memory, concentration), heart problems and even nasal cancer, especially by formaldehyde according to Ulker et al. [53]. Emissions of such VOCs in this study were relatively low. In general, particleboards containing PLB in the SL had lower TVOCs than those using PLB in the CL; this is mainly due to the fact that hot pressing assisted the evaporation of some TVOCs in the pressing procedure when PLB was applied in the SL. In contrast, the thick layers of the three-layer particleboards made it hard for the moisture to evaporate when PLB was used in the CL. 

## 4. Conclusions

A large amount of ethylene glycol as the liquefaction solvent remained in the PLB, and the bark after partial liquefaction did not experience notable chemical and structural changes as revealed by FTIR and SEM analysis. This indicated an initiation of liquefaction within 30 min, but a liquefaction can only be completed by prolonging the residence time over 30 min. PLB had significant effects on the physical and mechanical properties of the produced particleboards. The influence was higher when PLB was used in the CL than the SL regarding MOR and IB strength. However, the opposite was observed for the water-resistance ability of the PLB-based boards as examined by TS. It was also surprisingly found that the heat transfer rate of PLB was affected by the used amount of MUF adhesives. Tests on formaldehyde and VOCs emissions suggested the presence of unreacted ethylene glycol and its glycol esters, liberation of aldehydes from MUF and bark or wood and the formation of carboxylic acids in both the partial liquefaction process and particleboard production. 

The partial liquefaction method introduced by this paper provides new possibilities for exploring the potential use of lignocellulosic materials in particleboard production, especially those derived from industrial wastes/side-streams. The whole process from partial liquefaction until panel production produced no residues. This study showed some limitations when using PLB in three-layer particleboards associated with heat transfer issues during the pressing of such thick products as compared with single-layer particleboards. Thick boards and short pressing times make it hard for the heat to be transferred in the CL to activate the reaction between PLB and wood particles in order to provide bonds. In addition, PLB leads to a considerable increase in the emissions of VOCs. As a conclusion, PLB suits better for producing thinner boards such as single-layer particleboards or high-density fiberboards. Coating for such boards should be necessary for lowering the negative effects of the VOCs in PLB or possibly caused by PLB.

## Figures and Tables

**Figure 1 materials-16-01855-f001:**
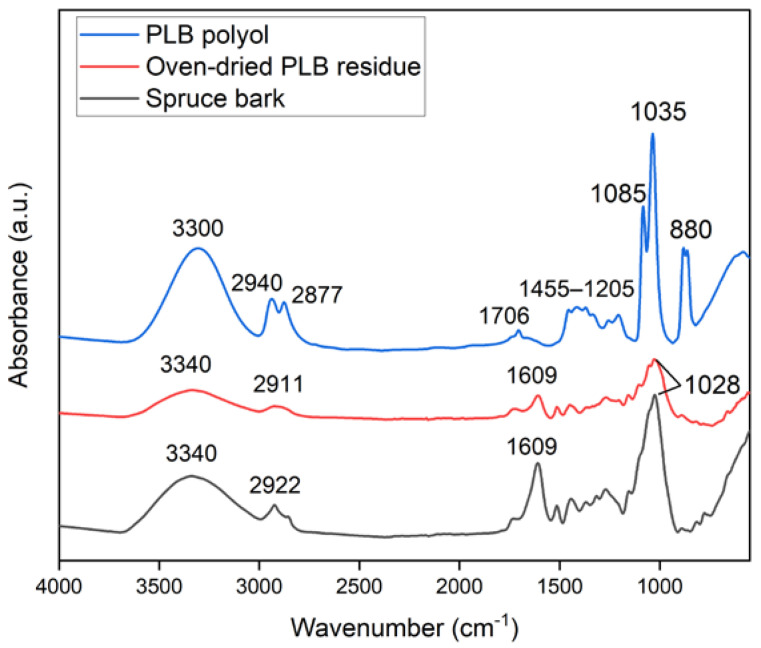
FTIR spectra of spruce bark, oven-dried PLB residue and PLB polyol.

**Figure 2 materials-16-01855-f002:**
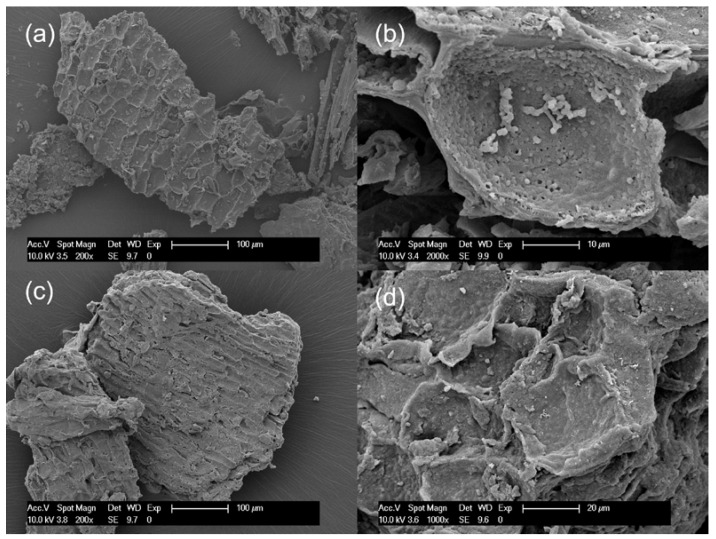
Scanning Electron Microscopy (SEM) micrographs: (**a**,**b**) raw spruce bark and (**c**,**d**) oven-dried PLB residues.

**Figure 3 materials-16-01855-f003:**
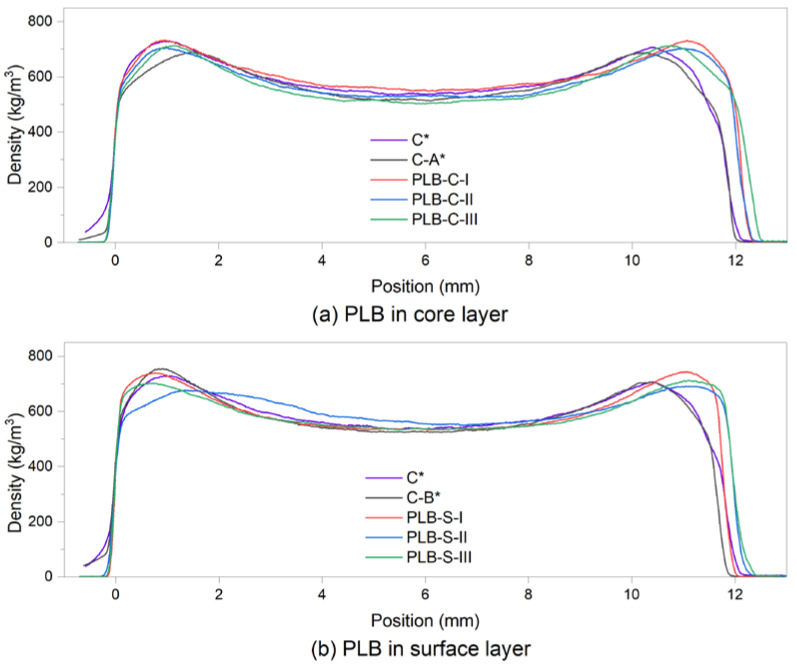
Vertical density profile of the particleboards at a target density of 620 kg/m^3^: (**a**) PLB was used in the core layers; (**b**) PLB was used in the surface layers. C*, C-A* and C-B* were reference boards.

**Figure 4 materials-16-01855-f004:**
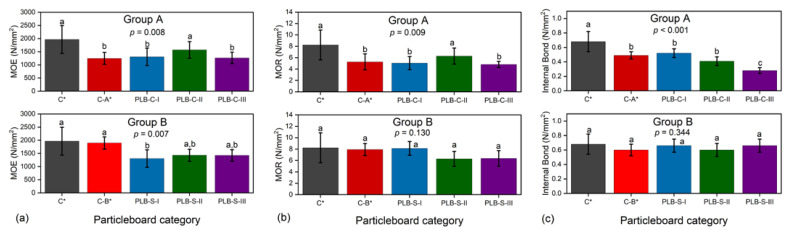
Average MOE (**a**), MOR (**b**), IB (**c**) of particleboards containing PLB in the surface or core layers. Values labelled with the same letter are not statistically different from each other (ANOVA, Tukey test, *p* < 0.05). Error bars represent standard deviations. C*, C-A* and C-B* were reference boards.

**Figure 5 materials-16-01855-f005:**
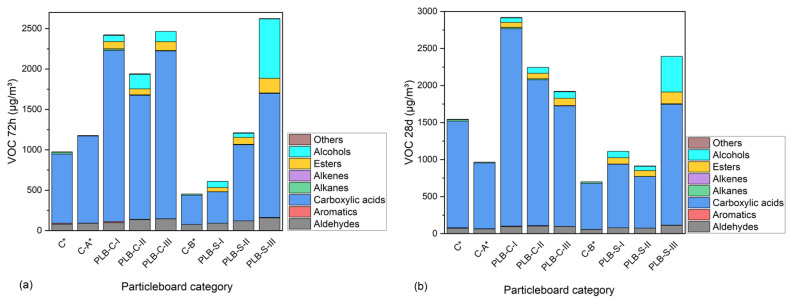
Concentrations of VOCs emitted from the particleboards after 72 h (**a**) and 28 days (**b**) of tests. C*, C-A* and C-B* were reference boards.

**Table 1 materials-16-01855-t001:** Parameters for preparing reference particleboards* and particleboards from groups A and B (PLB used in the core and surface layers, respectively). Duplicates were made for each panel type.

Parameter	Control	Group A				Group B			
Particleboard Type	C*	C-A*	PLB-C-I	PLB-C-II	PLB-C-III	C-B*	PLB-S-I	PLB-S-II	PLB-S-III
Surface layer (SL)-to-core layer (CL) ratio	0.4:0.6								
SL content	100% wood	100% wood				95% wood	90.1% wood + 9.1% PLB		
CL content	100% wood	95% wood	90.1% wood + 9.1% PLB			100% wood			
MUF content in SL (%)	12	12				12	12	10	8
MUF content in CL (%)	12	12	12	10	8			12	
Hardener content in SL (%)	3	3				3			
Hardener content in CL (%)	3	3				3			
Press temperature (°C)	210								
Pressing time (min)	3								
Dimensions (mm × mm × mm)	450 × 450 × 12								

**Table 2 materials-16-01855-t002:** Thickness and density of the particleboards. Group A contained PLB in the core layers and group B contained PLB in the surface layers. Values followed by a different letter within a row are statistically different (ANOVA and Tukey test, *p* < 0.05). Reference boards are marked with *.

Group		A						B					
Board Type		C*	C-A*	PLB-C-I	PLB-C-II	PLB-C-III	*p*	C*	C-B*	PLB-S-I	PLB-S-II	PLB-S-III	*p*
Thickness (mm)	Mean (SD)	12.11 (0.08) ^a,b^	11.98 (0.12) ^a^	12.29 (0.08) ^c^	12.20 (0.07) ^b,c^	12.30 (0.16) ^c^	< 0.001	12.11 (0.08) ^a^	11.83 (0.10) ^b^	11.88 (0.04) ^b^	12.17 (0.12) ^a^	12.14 (0.13) ^a^	< 0.001
Density (kg/m^3^)	Mean (SD)	612.1 (32.3) ^a^	599.3 (33.4) ^a^	623.9 (31.9) ^a^	609.1 (43.3) ^a^	566.2 (58.4) ^a^	0.082	612.1 (32.3) ^a^	613.7 (38.0) ^a^	645.2 (17.3) ^a^	641.6 (46.3) ^a^	624.6 (34.3) ^a^	0.208

**Table 3 materials-16-01855-t003:** Mean maximum density in the left and right sides, minimum density and medium density of 8 specimens for each particleboard type. C*, C-A* and C-B* were reference boards.

Group		A				B			
Board Type	C*	C-A*	PLB-C-I	PLB-C-II	PLB-C-III	C-B*	PLB-S-I	PLB-S-II	PLB-S-III
Max. density left (kg/m^3^)	729	689	732	705	713	755	740	676	703
Max. density right (kg/m^3^)	707	688	731	702	713	706	745	691	713
Min. density(kg/m^3^)	535	513	548	526	502	524	533	549	533
Medium density(kg/m^3^)	604	587	615	592	581	596	605	609	597

**Table 4 materials-16-01855-t004:** Results of 2 h and 24 h thickness swelling (TS) and water absorption (WA) of particleboards. C*, C-A* and C-B* were reference boards.

Group	Variant	TS 2 h (%)		TS 24 h (%)		WA 2 h (%)		WA 24 h (%)	
		Avg.	St. Dev.	Avg.	St. Dev.	Avg.	St. Dev.	Avg.	St. Dev.
A	C*	19.59 ^a^	2.05	23.78 ^a^	3.06	87.49 ^a^	11.87	98.72 ^a^	13.19
	C-A*	20.99 ^a^	2.59	25.02 ^a^	2.68	97.70 ^a^	5.88	107.69 ^a,b^	4.30
	PLB-C-I	19.35 ^a^	2.52	23.11 ^a^	3.58	89.07 ^a^	8.92	97.82 ^a^	7.99
	PLB-C-II	19.95 ^a^	2.18	24.15 ^a^	2.52	95.77 ^a^	3.63	105.42 ^a^	4.06
	PLB-C-III	21.82 ^a^	2.54	25.50 ^a^	2.58	110.18 ^b^	9.79	120.76 ^b^	12.07
	*p*	0.219		0.494		<0.001		<0.001	
B	C*	19.59 ^a,b^	2.05	23.78 ^a,b^	3.06	87.49 ^a^	11.87	98.72 ^a^	13.19
	C-B*	16.49 ^a^	0.93	19.69 ^a^	1.51	80.77 ^a^	11.24	94.30 ^a^	11.35
	PLB-S-I	17.62 ^a,b^	1.53	20.57 ^a,b^	2.19	77.28 ^a^	2.13	86.90 ^a^	1.77
	PLB-S-II	23.92 ^c^	3.61	28.34 ^c^	4.67	85.73 ^a^	7.39	97.33 ^a^	13.53
	PLB-S-III	21.12 ^b,c^	3.09	24.62 ^b,c^	3.43	88.02 ^a^	5.69	96.68 ^a^	6.12
	*p*	<0.001		<0.001		0.067		0.179	

Values followed by a different letter within a column are statistically different (ANOVA and Tukey test, *p* < 0.05).

**Table 5 materials-16-01855-t005:** Formaldehyde emissions of particleboards prepared with wood particles and PLB and bonded with MUF adhesive. C*, C-A* and C-B* were reference boards.

Particleboard Type	C*	C-A*	PLB-C-I	PLB-C-II	PLB-C-III	C-B*	PLB-S-I	PLB-S-II	PLB-S-III	E1 Limit
Formaldehyde emission (mg/m^2^·h)	0.361	0.293	0.382	0.296	0.323	0.346	0.298	0.365	0.284	3.5

## Data Availability

The data presented in the study are available on request from the corresponding author.

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
