# Peer review of "Properties and Emissions of Three-Layer Particleboards Manufactured with Mixtures of Wood Chips and Partially Liquefied Bark"

_materials, 2023, doi:10.3390/ma16051855_

Round 1

Reviewer 1 Report

Dear Authors,
below, please find several remarks, which in my opinion should improve your valuable manuscript:
- line 44-45 - about "flexible dimensions" - this mental shortcut suggests that particleboards are like a rubber; I suggest saying "low cost and availability on the market in a wide range of dimensions"
- line 48 - regarding forest-based valorized resources, in the context of recycled wood, please consider citation https://doi.org/10.3390/ma15238487
- line 103-111 - please highlight the aim (goal) of your work at the end of this paragraph
- line 153 - "stubs coated with gold" suggest that you coated only the elements of the measuring stand, and not tested samples; suggest to say "on aluminium stubs, and were coated with gold"
- table 1, last two rows, seems like thickness is one of the dimensions, so why you excluded it from the last row? Also please correct the unit of thickness
- figure 3, both plots - please correct the density units (should be kg/m3)
- figure 4 b) - please correct the unit of MOR (should be N/mm2)

Best regards!

Author Response

Dear reviewer,

Thank you for your good comments and useful suggestions. Please find the response letter attached. 

Reviewer 2 Report

Please check my details comment in the attachment

Author Response

Dear reviewer,

Thank you for your valuable comments and suggestions on the statistical analysis. We have followed your suggestion. Please check our reponse to your comments in the attachment.

Best regards,

Authors

Reviewer 3 Report

Liquefied bark was successfully prepared as a replacement in particleboard manufacturing in this study. Several methods were used to compare and contrast the liquefied bark products. The concept of the work itself is timely. However, there are several serious issues that must be addressed before publication. Liquefied bark has been widely reported. The current work merely highlights the replacement of liquefied bark in the core and surface layer of three-layer particleboard. The concept's novelty appears to be limited. The authors failed to emphasise the investigation's key contributions. What's new? Under such circumstances, it is difficult to justify the acceptance of this paper.

1.       The revised manuscript should include the critical parameters of liquefied bark, such as Valorizations of tree barks - wooden panels, tannin, resins, and foams, as well as the thermochemical conversion of bark (via phenolysis, direct liquefaction, and pyrolysis), and applications of bark thermochemical conversion products.

2.       What functional groups are represented by 1035 and 1028 cm-1?

3.      Because this was expected, related mechanism comments should be provided to establish the relationship between low Formaldehyde emissions and liquefied bark replacement in particleboard.

Author Response

Dear reviewer,

We really appreciate your valuable comments. Please find our response letter attached. 

Best regards,

Authors

Reviewer 4 Report

The research paper has a very good scientific level. The content of the paper is valuable from the theoretical and practical point of view. The article presents a current topic. The aim of research is interesting and beneficial for this research field. The topic is convenient for the scope of the journal. The title of scientific article is clear and it sufficiently reflects content. The abstract and key words are informative. The figures and tables appropriately complement the presentation of the scientific work results. The manuscript is well organized.

1.    I recommend clearly and shortly emphasizing the main goal of the research in the Introduction.

2.       Statistical analysis is described very generally, so I recommend a more detailed description.

3.     I recommend highlighting the novelty and the benefits of the research in Conclusions.

4.       What is the main reason that bending properties did not correlate well with the density profile in this work?

5.       Lines: 310 – 311: Cit. ”All the particleboards produced in this work did not exceed the limit by European standard and had relatively lower formaldehyde emissions than the literature.” The meaning of the sentence is ambiguous. The EU Standards should by added. The end of sentence should be reformulated. The literature should be more specific.

GENERAL JUDGEMENT

The paper is acceptable for publication after minor revision.

Author Response

Dear reviewer,

Thanks for your good comments. Please kindly find our response letter attached.

Best regards,

Authors

Round 2

Reviewer 2 Report

No further comment.